# Femtosecond pulse amplification on a chip

Mahmoud A. Gaafar [1,7], Markus Ludwig[1,7], Kai Wang [2], Thibault Wildi [1], Thibault Voumard[1], Milan Sinobad [1], Jan Lorenzen [1], Henry Francis[3], Jose Carreira[3], Shuangyou Zhang[4], Toby Bi [4,5], Pascal Del'Haye [4,5], Michael Geiselmann[3], Neetesh Singh [1], Franz X. Kärtner [1,6], Sonia M. Garcia-Blanco[2] & Tobias Herr [1,6] ✉

Femtosecond laser pulses enable the synthesis of light across the electromagnetic spectrum and provide access to ultrafast phenomena in physics, biology, and chemistry. Chip-integration of femtosecond technology could revolutionize applications such as point-of-care diagnostics, bio-medical imaging, portable chemical sensing, or autonomous navigation. However, current chip-integrated pulse sources lack the required peak power, and on-chip amplification of femtosecond pulses has been an unresolved challenge. Here, addressing this challenge, we report >50-fold amplification of 1 GHz-repetition-rate chirped femtosecond pulses in a CMOS-compatible photonic chip to 800 W peak power with 116 fs pulse duration. This power level is 2–3 orders of magnitude higher compared to those in previously demonstrated on-chip pulse sources and can provide the power needed to address key applications. To achieve this, detrimental nonlinear effects are mitigated through all-normal dispersion, large mode-area and rare-earth-doped gain waveguides. These results offer a pathway to chip-integrated femtosecond technology with peak power levels characteristic of table-top sources.

Femtosecond laser pulses with high-peak power are a fundamental resource in photonics. By temporally concentrating laser light, they provide unique access to ultrashort timescales and nonlinear optical effects and have found applications across many disciplines, including bio-medical imaging[1–3], femtochemistry[4,5], optical precision spectroscopy, molecular sensing, low-noise signal generation and time keeping[6–8]. Recent developments of photonic-chip integration of femtosecond sources via microresonators[9–11], on-chip mode-locked lasers[12–15], and integrated electro-optic pulse generators[16] have created tremendous opportunities for femtosecond technology to become accessible at low-cost with high-volume, compact footprint and beyond specialized laboratories. These developments hold the potential for large-scale societal benefits such as advanced point-of-care diagnostics, distributed environmental monitoring, and mobile applications such as signal generation for navigation and communication. A current roadblock, however, is the low output peak power of chip-integrated femtosecond sources (around 1 W, or much below that), which is often too low for their envisioned application. Thus, almost all proof-of-concept demonstrations with ultrashort pulses from on-chip sources have critically relied on table-top optical amplifiers. Integrating these pulse amplifiers on a chip is a long-standing challenge.

Previous work has already demonstrated chip-integrated (quasi-) continuous-wave amplifiers based on rare-earth-doped waveguides[17–26] sustaining power levels up to 1 W[27], heterogeneous semiconductor integration[28–31] or nonlinear parametric gain[32–34]. However, the amplification of femtosecond pulses in an integrated photonics setting has remained challenging, with nonlinear pulse distortion occuring already at low peak power levels of 20 W[25]. Indeed, the strong light confinement in integrated waveguides results in strong nonlinearities,

[1]Deutsches Elektronen-Synchrotron DESY, Notkestr. 85, 22607 Hamburg, Germany. [2]Integrated Optical Systems, MESA+ Institute for Nanotechnology, University of Twente, 7500AE Enschede, The Netherlands. [3]LIGENTEC SA, EPFL Innovation Park, Chemin de la Dent-d'Oche 1B, Switzerland CH-1024 Ecublens, Switzerland. [4]Max-Planck Institute for the Science of Light, 91058 Erlangen, Staudtstr. 2, Germany. [5]Department of Physics, FAU Erlangen-Nürnberg, 91058 Erlangen, Germany. [6]Department of Physics, Universität Hamburg, Luruper Chaussee 149, 22761 Hamburg, Germany. [7]These authors contributed equally: Mahmoud A. Gaafar, Markus Ludwig. ✉e-mail: tobias.herr@desy.de

typically 1000 times higher than those in optical fibers. While often advantageous[35], e.g., for broadband supercontinua[36–39], nonlinear optical effects during the amplification process can lead to the irreversible degradation of the pulse within sub-mm propagation distances.

Table-top laser systems have historically faced a similar challenge. Nonlinear optical effects would prevent the amplification of pulses or even result in damage of the amplifier itself[40]. A major breakthrough came through the demonstration of *chirped pulse amplification* (CPA)[41]. In CPA, the pulse is temporally stretched (*chirped*) to lower its peak power, amplified, and then temporally re-compressed only after amplification. While stretching can be readily accomplished at low power levels through a dispersive optical element, amplification and compression require advanced optical setups to manage nonlinearity, group velocity dispersion (GVD), and for ultrashort pulses, also higher-order dispersion.

Here, we demonstrate femtosecond pulse amplification on an integrated photonic chip to hundreds of Watts of peak power (Fig. 1a), a previously lacking cornerstone for chip-scale femtosecond technology. Our approach pursues a concept that resembles CPA to mitigate the strong nonlinear optical effects that are usually associated with integrated photonics. Pre-chirped input pulses propagating through the amplifier are simultaneously amplified and temporally compressed by managing nonlinearity and dispersion so that ultrashort pulse duration and high-pulse peak power are reached at the amplifier's output facet (Fig. 1b). In this way, we achieve >50-fold amplification of ultrashort pulses from a 1 GHz femtosecond source (center wavelength 1815 nm) to 800 W peak power and a final pulse duration of 116 fs. Our experimental results are in excellent agreement with a numerical pulse propagation model that includes the dynamics of the Tm-doped gain medium.

## Results

The amplifier (Fig. 1c) is fabricated in a CMOS-compatible, scalable silicon nitride ($Si_3N_4$) photonic-chip platform (here with an 800-nm-thick waveguide layer). The total length of the amplifier waveguide structure is 12 cm, which is integrated in a 15 mm²-scale footprint. It comprises a total of ~10 cm of straight waveguide sections that provide gain and curved waveguide sections for compact integration. To achieve on-chip gain, the silica ($SiO_2$) top cladding is selectively removed in the amplifier section down to 200 nm above the $Si_3N_4$ layer. In the next step, a 1000-nm-thick thulium-doped alumina ($Tm^{3+}:Al_2O_3$) gain layer is deposited. The refractive index of the gain layer $n \approx 1.72$ is between that of $Si_3N_4$ and $SiO_2$. The estimated Tm concentration is $3.5 \times 10^{20}$ cm⁻³. A 1000-nm-thick silica layer is deposited on top for protection (see "Methods" for more details on fabrication). In the straight gain sections, the width of the $Si_3N_4$ waveguide is reduced to 300 nm, resulting in a single-mode waveguide with large mode-area that is mostly contained in the active alumina gain layer (Fig. 1d). Such large mode-area (LMA) gain waveguides[14] can provide high gain per unit length, low-nonlinearity, and high saturation power[27], and have already enabled Q-switched lasers with nanosecond pulse duration[42,43]. In the curved sections, the waveguide width is 1000 nm for strong mode confinement (Fig. 1e) and low-loss propagation; Euler-bends with a minimal radius of 200 μm guarantee an adiabatic low-loss transition to the bends[44].

The amplifying and bending waveguide sections are connected by straight adiabatically tapered waveguides. To increase mode confinement for low reflections and scattering at the cladding material transition, input and output waveguides are 2000 nm wide. Inverse tapers at the facets of the chip enable low-loss coupling from and to optical fibers or free space. The footprint of the entire amplifier structure is below 15 mm². The choice of Tm-doping is motivated by its large gain bandwidth[21,42,43,45,46] and its importance for emerging applications such as laser ranging, free-space communication[47] and mid-infrared supercontinua[48]; however, the amplifier platform can also support erbium-doping for operation in the telecommunication wavelength window[49], and likely other rare-earth dopants such as neodymium and ytterbium for operation around 1 μm wavelength.

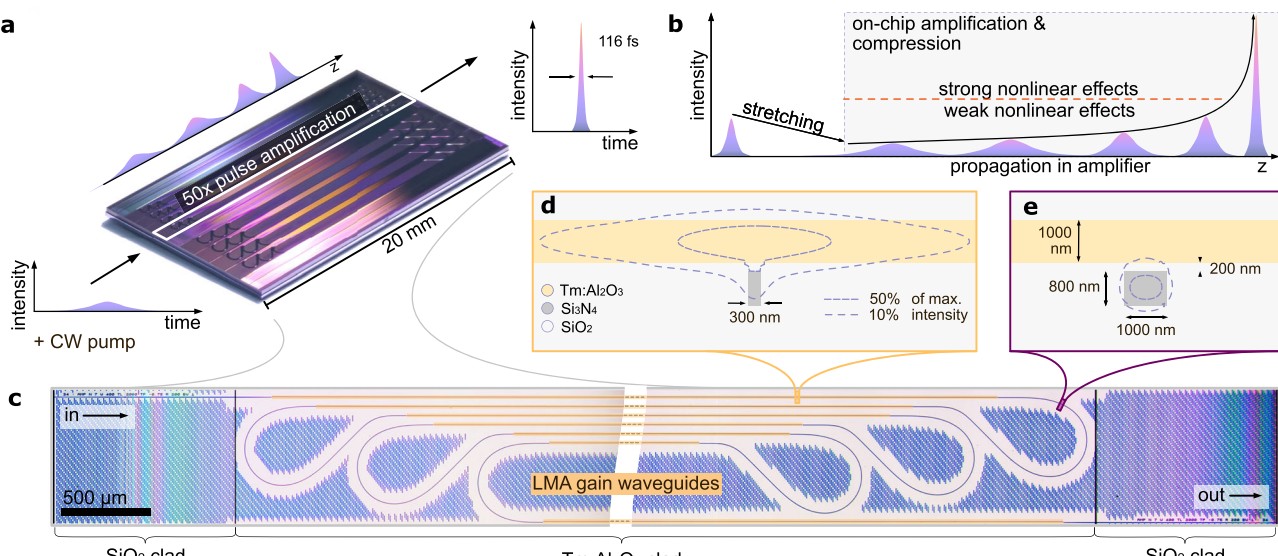

**Fig. 1 | Ultrafast pulse amplification on a chip. a** Photograph of the chip, hosting several amplifiers. Low-energy chirped pulses are amplified 50-fold. The output pulse is a high-peak power, nearly time-bandwidth-limited femtosecond pulse. **b** After temporally stretching (chirping) the input pulse, it is coupled to the amplifier chip, where it is gradually amplified and compressed, avoiding strong nonlinearities until passing the output section. **c** Composite photograph of an amplifier structure consisting of large mode-area (LMA) gain waveguides folded into a small 15 mm²-scale footprint. The waveguides are defined in a silicon nitride ($Si_3N_4$) layer, where within the area of the gain waveguides, an active thulium-doped alumina layer ($Tm^{3+}:Al_2O_3$) is used as a top cladding; in the input and output sections, a silica ($SiO_2$) cladding is used. **d** Layer structure in the LMA gain waveguide section. Intensity contours of the optical mode illustrate that most of the power is propagating in the gain layer for optimal amplification. **e** Same as (**d**) but for the bent waveguide sections. The mode is strongly confined to the $Si_3N_4$-core, enabling low-loss propagation through the bends.

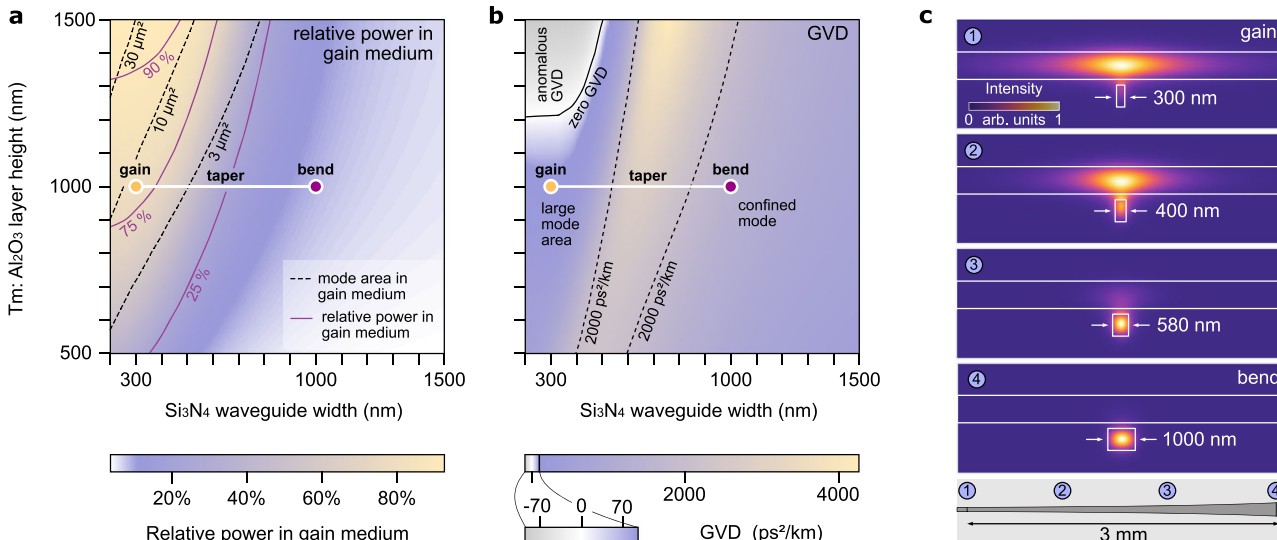

**Fig. 2 | Amplifier design. a** Relative power fraction in the transverse magnetic (TM) mode that is contained in the $Tm^{3+}$:$Al_2O_3$ gain layer (colorscale, purple contours) and mode-area in the gain layer (dashed black contours) in dependence of the $Tm^{3+}$:$Al_2O_3$ layer height and the $Si_3N_4$ waveguide width; cf. "Methods" for definition of the mode-area. The locations of the gain waveguides ("gain"), bends ("bend") and tapers ("taper") in the parameter space are indicated. **b** Group velocity dispersion (GVD) (colorscale, dashed black contours); axes and labels as in (**a**). **c** Simulated intensity profiles for different $Si_3N_4$ waveguide widths as they occur in gain, bent and tapered waveguide sections. Their positions along the adiabatic taper-profile are indicated. The simulations are based on a finite-element method model.

The design of the geometry of the amplifier is based on numerical simulation of key waveguide parameters obtained through a finite-element electromagnetic mode solver. For these simulations to be accurate, we utilize wavelength-dependent refractive index data that was obtained for all involved materials through broadband ellipsometry. Figure 2a, b shows, for the fundamental transverse magnetic mode (TM), the relative optical power in the gain layer and the GVD as a function of the $Si_3N_4$ waveguide width and the $Tm^{3+}$:$Al_2O_3$ layer height. To achieve high gain and high saturation power, a large power fraction of the optical mode in the $Tm^{3+}$:$Al_2O_3$ layer is generally desirable. This also reduces the impact of nonlinear optical effects by (1) lowering the intensity through a large mode-area, (2) a 10-fold reduced nonlinear material index of the gain layer compared to the silicon nitride waveguide, and (3) by achieving high-gain amplification over short distances. However, a too large mode-area (and the correspondingly high number of excited ions) would increase the minimum signal input power required to avoid excessive spontaneous emission or parasitic lasing, particularly in a co-propagating (forward) pumping scheme. In addition to a high power fraction in the gain medium, we aim at an all-normal GVD along the entire amplifier structure to achieve monotonous temporal pulse compression and robustness against pulse breakup by nonlinear modulation instability (MI)[50]. As Fig. 2a and b show, it is indeed possible to simultaneously fulfill these criteria for the gain-waveguide, taper and bend for a $Tm^{3+}$:$Al_2O_3$ layer thickness below 1200 nm. Here, we use a chip with a layer thickness of 1000 nm in the normal GVD regime.

The location of the gain waveguide, bends and connecting tapers in the design parameter space are indicated in Fig. 2a, b, and their mode profiles are shown in Fig. 2c. In this design, the effective mode-area in the gain layer is $A_{eff}^{gain} \approx 7 \mu m^2$ with an effective nonlinear parameter of $\gamma = 0.007$ $W^{-1}m^{-1}$ (in the 1000 nm wide waveguide $\gamma = 0.7$ $W^{-1}m^{-1}$) (see "Methods"). The group-delay dispersion (GDD, i.e., the length-integrated GVD) of the amplifier chip is $7.79 \times 10^{-26}$ $s^2$, which is large enough to allow pre-chirping to picosecond duration pulses. For ultrashort pulses, besides second-order dispersion (GVD), third-order dispersion can be important. As we show in the Supplementary information (Section 1, Fig. S1), the distinct third-order dispersion

characteristics of bends and tapers can be leveraged to minimize the detrimental impacts of third-order dispersion.

The experimental setup for testing the amplifier is shown in Fig. 3a, including the input signal pulse and the 1610 nm continuous-wave (CW) pump sources that are externally combined and coupled to the chip. For convenience, the pump source is here implemented via a diode laser that is amplified in an erbium-doped fiber amplifier. Future integration may utilize high-power single-mode laser diodes, which are commercially available with power levels of larger than 400 mW, and which could be combined in a polarization multiplexed and/or bi-directional pumping configuration. Alternatively, 790 nm pump diodes may be used to excite the gain medium via power-efficient cross-relaxation processes; given the significance of Tm-doped gain materials, we anticipate high-power pump diodes to become readily available. The output of the amplifier is collected via free-space optics to avoid additional dispersive or nonlinear optical effects and is characterized via frequency-resolved optical gating (FROG)[51]. As input pulse source, we use a femtosecond laser source (cf. "Methods") providing close to bandwidth-limited pulses of 80 fs duration at a central wavelength of 1815 nm and a pulse repetition rate of 1 GHz of which we couple 1.81 mW of average power (1.81 pJ pulse energy) to the chip. Prior to coupling the pulses to the chip, they are pre-chirped in a short stretch of standard anomalous dispersion optical fiber. The fiber can easily be adjusted in length to explore different levels of pre-chirping. Such pre-chirping could also be implemented on-chip without suffering from nonlinear effects owing to the low pulse energy prior to amplification. Relevant approaches include strongly anomalous dispersion waveguides[16], Bragg-gratings[52–54], or coupled gratings[55,56].

As an aside, we note that it is, in principle, also possible to send the pulses directly, without pre-chirping, into the amplifier and to perform re-compression after the amplification (Supplementary information, Section 2, Fig. S2). However, this would require dispersion compensation at higher power levels after amplification, which, in contrast to pre-chirping at lower power, is challenging due to non-trivial nonlinear compression dynamics. If carefully optimized, self-similar nonlinear pulse amplification in the normal dispersion regime[57,58], followed by anomalous dispersion and/or nonlinear pulse compression, may result

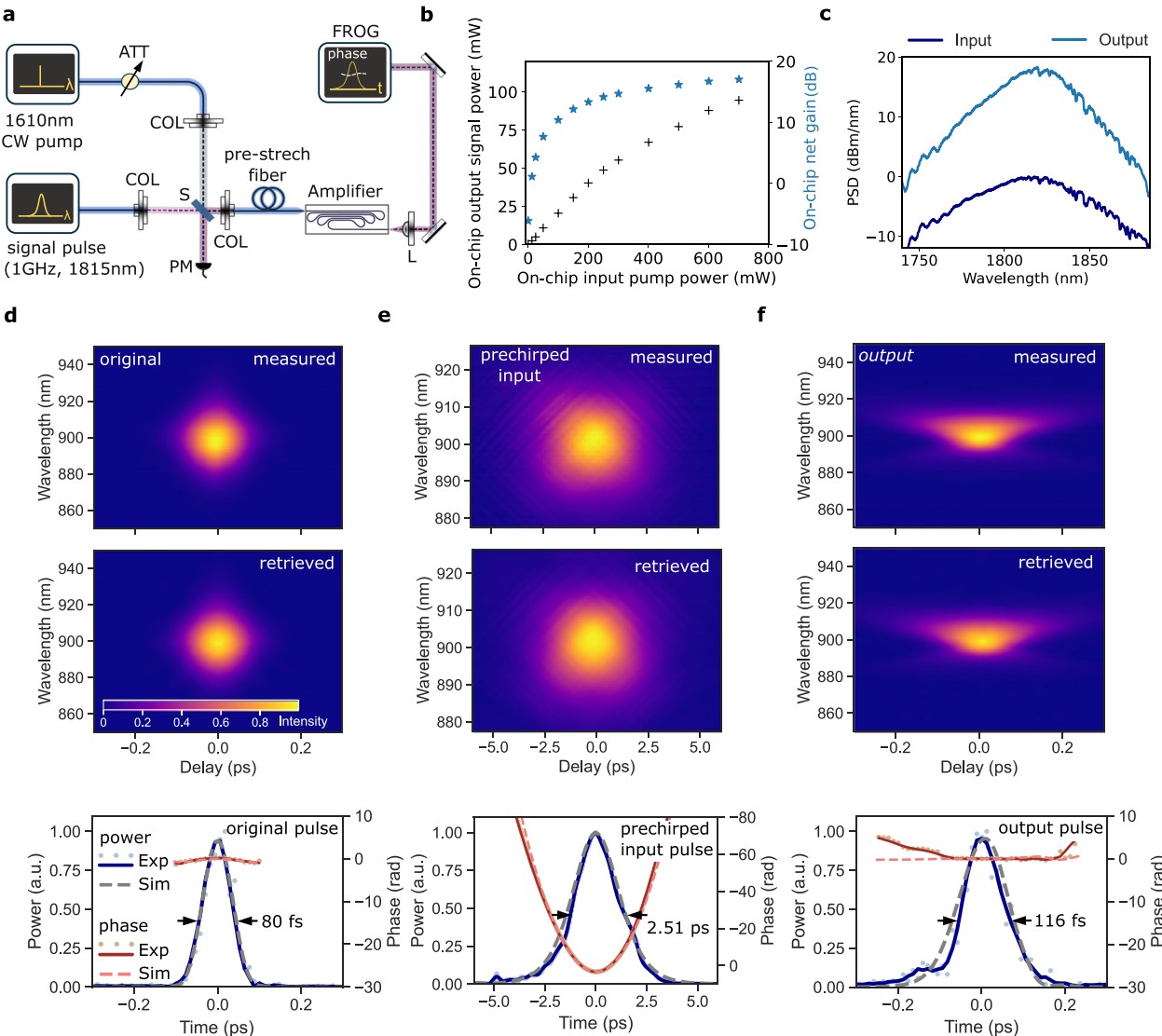

**Fig. 3 | Experiments. a** Experimental setup. CW: continuous-wave; ATT: attenuator; COL: collimator; S: beam-splitter; L: lens; PM: power-meter. **b** Amplifier output power and on-chip net gain as function of the on-chip pump power. **c** Optical spectra before and after amplification; PSD: power spectral density. **d**–**f** Frequency resolved optical gating (FROG) traces showing second harmonic intensity as function of FROG delay (256 × 256 data points) and reconstructed temporal pulse (power and phase) of the original, the pre-chirped input and the output pulses. Dots and solid lines represent measured and smoothed measured data, dashed lines represent simulated data. The respective FROG errors are $1.282 \times 10^{-5}$, $2.75 \times 10^{-6}$, and $1.19 \times 10^{-5}$.

in pulse shortening; however, this is not pursued here and only works for very specific input pulse parameters.

Figure 3b shows the average power of the amplified pulses as well as the on-chip net gain for different levels of pump power. For the highest on-chip pump power of 700 mW, we observe an on-chip output signal power of 95 mW and a maximal gain of 17 dB, i.e., a factor of 52 (see "Methods" for details on calibration); the optical spectrum before and after amplification is shown in Fig. 3c. Next, we vary the length of the pre-chirping fiber and measure the temporal characteristics of the output pulses directly after the chip via FROG. For an optimal pre-chirping fiber length of 146.5 cm, commensurate with the opposite amount of total dispersion along the amplifier chip, we obtain an amplified and compressed pulse directly at the output facet of the chip of 116 fs duration. The temporal characterization of the original, pre-chirped and amplified pulses are shown in Fig. 3d, e and f, respectively. The pre-chirped pulses have a duration of 2.5 ps and a markedly quadratic temporal phase. In contrast, the output pulses of the amplifier are 116 fs in duration and exhibit a largely flat

temporal phase indicating that they are close to time-bandwidth-limited. The asymmetric low-intensity tail of the reconstructed pulse (and a 'butterfly' shape in the FROG trace) hints at a weak residual third-order dispersion. In total, this experiment signifies the successful more than 50-fold amplification of a femtosecond pulse in an integrated photonic chip. The obtained on-chip pulse energy of 95 pJ and peak power of >800 W is well-suited for photonic chip-based spectral generation from infrared to ultraviolet wavelength[59,60] or self-referencing[61] for absolute optical metrology. In integrated microwave photonics[62], ultrashort pulses could improve shot-noise limited performance by several orders of magnitude[63] with impact on navigation and communication applications. If higher average output power is needed, a longer amplifier structure and larger mode-area gain waveguides can be used. We note that the final output pulses of 116 fs duration are longer than the 80 fs input pulse. This correlates well with their respective spectral bandwidth of 5.1 THz and 3.5 THz, respectively. We, therefore, attribute the increase in pulse duration to a finite gain bandwidth, but also to residual

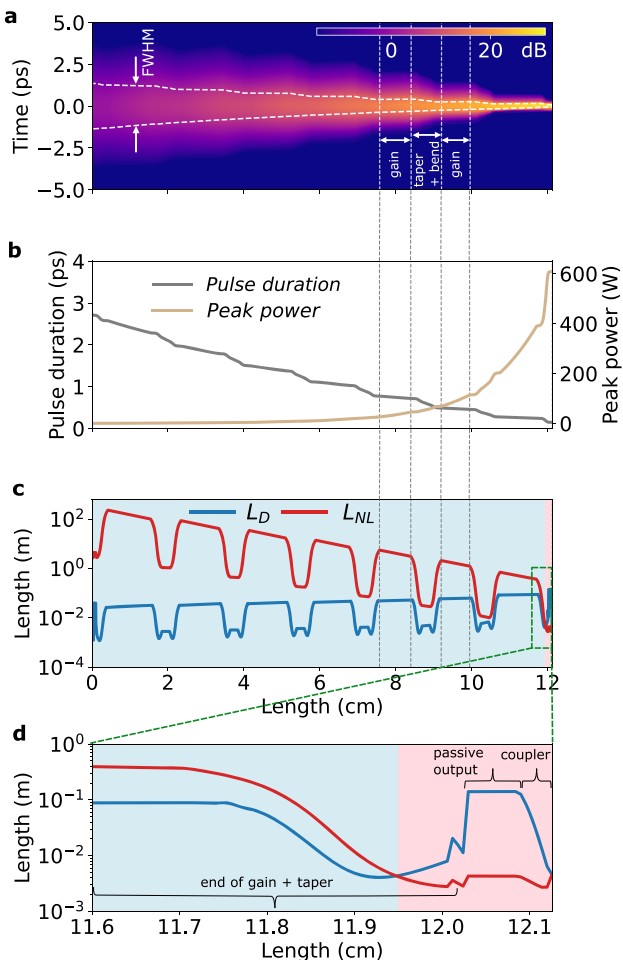

**Fig. 4 | Numerical simulation. a** Evolution of the temporal pulse power while propagating through the amplifier chip in a co-moving reference frame. One sequence of the alternating gain, bend and taper sections is indicated. The contours indicate the full-width-half-maximum (FWHM) pulse duration. **b** Pulse duration and pulse peak power while propagating through the amplifier chip. **c** Evolution of the pulse's dispersion length $L_D$ and nonlinear length $L_{NL}$ while propagating through the amplifier chip. The blue background color highlights where the propagation is dominated by linear optical effects ($L_D < L_{NL}$); nonlinear optical effects dominate only in the last 1.5 mm of the entire >12 cm long propagation distance (red background color). **d** Magnified portion of panel (**c**), where the gain section and taper to the output waveguide, the passive output waveguide, and the inverse taper for chip-output coupling are indicated. See "Methods" for details on the simulation.

third-order dispersion. Potentially, although less likely, nonlinear spectral compression could play a role[64].

Complementing the experiment, we perform numerical simulations of on-chip pulse amplification based on the nonlinear Schrödinger equation (NLSE) combined with a thulium gain model (see "Methods"). Using the experimental parameters of the input pulse, pump power and waveguide parameters, we simulate the pulse amplification and compare it with the experimental data in Fig. 3d–f (bottom). The simulation is in excellent agreement with the experimental observations and we can use it to gain further insights into the pulse propagation during the amplification process. Figure 4a shows the simulated temporal dynamics of amplification and pulse shortening. We extract the pulse duration as a function of propagation distance and confirm the intended monotonous pulse compression that permits to keep the pulse peak power $P_p$ and the impact of nonlinear effects low throughout most of the amplifier length (Fig. 4b). To illustrate this point, we plot in Fig. 4c the

nonlinear length $L_{NL} = (\gamma P_p)^{-1}$, the characteristic propagation length scale on which nonlinear effects become relevant. It can be seen that $L_{NL}$ is consistently longer than the stretches of propagation to which it applies, indicating the limited impact of nonlinear effects. We also indicate the characteristic length scale of dispersive effects $L_D = TBP^2/(\Delta f^2 |GVD|)$, which is significantly shorter than $L_{NL}$ for most of the propagation, confirming that pulse propagation is dominated by linear optical effects ($\Delta f$ is the spectral bandwidth of the pulse as obtained from the simulation data at each point of the propagation and TBP is the time-bandwidth product, here set to 0.315).

## Discussion

In summary, we have demonstrated ultrashort femtosecond pulse amplification in an integrated photonic chip. The central challenge of strong optical nonlinearity inherent to integrated photonics is effectively addressed through tailored large mode-area gain waveguides and the design of the waveguide's dispersion, permitting stable pulse propagation and compression, including in the tightly confining bends. The achieved 50-fold on-chip amplification leads to a peak power of 800 W. Longer amplifying waveguides and larger mode-area can support even higher average output power, accommodating higher pulse repetition rates. As we discuss in the Supplementary information (Section 3, Fig. S3), substantially increasing the peak power while avoiding nonlinear effects would require a design with increased waveguide dispersion or shorter pulses, which are, in principle, supported by the Tm-doped gain medium. Future work may also leverage deliberate nonlinear effects such as self-similar pulse amplification in the normal dispersion regime[57,58] followed by anomalous dispersion and potentially nonlinear compression to provide high-power few-cycle pulses[65]. As such, our results open new opportunities for scalable high-power fully-chip-integrated ultrafast sources. More broadly, they bring femtosecond technology with pulse peak power otherwise only attainable in table-top systems to the chip level.

## Methods

### Sample fabrication

The passive $Si_3N_4$ chips were fabricated by Ligentec SA using UV optical lithography based on an 800-nm-thick $Si_3N_4$ waveguide platform embedded in a silica cladding. In the area of the gain waveguides, the top silica cladding is removed (a 200 nm silica spacer layer remains). A $Tm^{3+}$: $Al_2O_3$ gain layer with a thickness of 1000 nm is then deposited in the local opening using radio frequency sputtering technique[66] with an estimated $Tm^{3+}$ ion concentration of $3.5 \times 10^{20}$ cm$^{-3}$. The gain layer is cladded with a 1000-nm-thick silica layer on top for protection. Finally, the chip is annealed at 500 °C for 6 h to reduce the OH and water absorption in the cladding layer.

### Nonlinear coefficient and effective mode-area

The nonlinear coefficient $\gamma$ of the waveguides is[67]

$$\gamma = \frac{\epsilon_0 \omega_s \iint n^2(x,y) n_2(x,y) |\mathbf{E}(\mathbf{x},\mathbf{y})|^4 \mathrm{d}x\mathrm{d}y}{\mu_0 c \; |\iint \mathrm{Re}[\mathbf{E}(x,y) \times \mathbf{H}^*(x,y)]_z \mathrm{d}x\mathrm{d}y|^2}$$

where the integral extends over the entire heterogeneous material geometry. $\mathbf{E}$ and $\mathbf{H}$ are the complex electric and magnetic field vectors, $x$ and $y$ the transverse spatial coordinates. The effective mode-area $A_{eff}^{gain}$ in the homogeneous gain medium is

$$A_{eff}^{gain} = \frac{\left( \iint_{gain} |\mathbf{E}(x,y)|^2 \mathrm{d}x\mathrm{d}y \right)^2}{\iint_{gain} |\mathbf{E}(x,y)|^4 \mathrm{d}x\mathrm{d}y}$$

where the integration is restricted to the $Tm^{3+}$:$Al_2O_3$ gain layer.

## Ultrafast signal source

Pulses of 150 fs in duration, a repetition rate of 1 GHz and a central wavelength of 1560 nm are launched into a highly nonlinear fiber (HNLF) to generate a Raman-shifted soliton pulse centered at a wavelength of 1815 nm. A free-space 1700 nm long-pass filter (LPF) is used to remove residual short wavelength contributions.

## Power calibration and gain measurements

Pump and signal fiber-to-chip coupling losses are measured using passive (silica gladding, no gain layer) waveguides and a symmetric lensed fibers configuration for input and output coupling. Measured pump and signal coupling losses are 2.9 and 4.2 dB per facet, respectively. We extract the net gain by directly comparing the output and input signal powers after calibrations and after filtering out the transmitted pump using a free-space long-pass filter.

## Model for the amplification dynamics

Based on the NLSE, our model for propagation and amplification of the signal pulses combines effects of nonlinearity, gain and dispersion. The optical nonlinearity is treated in the time domain

$$\frac{\partial A_k(z)}{\partial z} = i\gamma |A_k(z)|^2 A_k(z),$$

where $A_k(z)$ is the complex temporal field amplitude of the signal, defined in the co-moving time frame on the discrete temporal points $T_k$ ($k = 0, 1, 2, \ldots, N-1$) and normalized such that $\sum_k |A_k(z)|^2 \Delta T$ is the pulse energy at position $z$ with $\Delta T = T_{k+1} - T_k$. Dispersion and gain are treated in the frequency domain

$$\frac{\partial \tilde{A}_j(z)}{\partial z} = \frac{g_{S,j}(z)}{2}\tilde{A}_j(z) + i\sum_{n \geq 2}\frac{\beta_n}{n!}(\omega_j - \omega_S)^n \tilde{A}_j(z)$$

where $\tilde{A}_j(z)$ is the discrete Fourier-transform of $A_k(z)$, with frequencies $\omega_j$ ($j = 0, 1, 2, \ldots, N-1$), and $\omega_S$ is the signal's center frequency. The frequency resolution is $\Delta\omega = \frac{2\pi}{N\Delta T}$, the signal gain is $g_{S,j}(z)$, and the $\beta_n$ describes the dispersion.

We model the optical gain by accounting for the populations in the levels ($^3H_6$, $^3F_4$ and $^3H_4$). The gain model also accounts for the fractions $f_a$ and $f_q$ of active and quenched ions ($f_a + f_q = 1$), respectively. Assuming a steady state for the gain medium, the rate equations, including energy transfer upconversion (ETU) and cross-relaxation (CR) processes for 1610 nm pumping, are given by[18,68,69]:

$$\frac{dN_{2,a/q}(z)}{dt} = W_{ETU1}N_{1,a/q}^2(z) - W_{CR}N_{0,a/q}N_{2,a/q} - \frac{1}{\tau_2}N_{2,a/q}(z) = 0$$

$$\frac{dN_{1,a/q}(z)}{dt} = R_{P,a/q}(z) - R_{S,a/q}(z) - (2W_{ETU1} + W_{ETU2})N_{1,a/q}^2(z)$$
$$+ 2W_{CR}N_{0,a/q}N_{2,a/q} + \frac{1}{\tau_2}N_{2,a/q}(z) - \frac{1}{\tau_{1,a/q}}N_{1,a/q}(z) = 0$$

$$f_{a/q}(z)N_d = N_{0,a/q}(z) + N_{1,a/q}(z) + N_{2,a/q}(z)$$

$$R_{P,a/q}(z) = \frac{1}{\hbar\omega_P}\frac{P_P(z)}{A_{\text{eff}}(\omega_P)} \times \Big[\sigma_a(\omega_P)N_{0,a/q}(z) - \sigma_e(\omega_P)N_{1,a/q}(z)\Big]$$

$$R_{S,a/q}(z) = \sum_j \frac{1}{\hbar\omega_S}\frac{P_{S,j}(z)}{A_{\text{eff}}(\omega_S)} \times \Big[\sigma_e(\omega_j)N_{1,a/q}(z) - \sigma_a(\omega_j)N_{0,a/q}(z)\Big]$$

where the concentrations $N_{0,1,2}$ describe the populations in the levels $^3H_6$, $^3F_4$ and $^3H_4$, respectively, the subscripts $a/q$ denote the active or quenched ions and $N_d$ is the total ion concentration; $\tau_i$ is the luminescence lifetime of level $i$ and $W_{ETU1}$ and $W_{ETU2}$ are parameters accounting for the ETU processes $^3F_4, ^3F_4 \rightarrow ^3H_4, ^3H_6$ and $^3F_4, ^3F_4 \rightarrow ^3H_5, ^3H_6$ and $W_{CR}$ describes the CR process $^3H_4, ^3H_6 \rightarrow ^3F_4, ^3F_4$[69]. In our simulation, we did not observe a significant effect from the ETU processes with values as in ref. [69]. We thus neglect CR and also set $f_q = 0$. $R_{p,a/q}$ and $R_{s,a/q}$ are the pump and signal rates. The pump power $P_P$ is monochromatic with frequency $\omega_P$ and the signal power contained in

the spectral discretization intervals is $P_{S,j}(z) = \frac{\Delta T^2}{T}|\tilde{A}_j(z)|^2\frac{\Delta\omega}{2\pi} = N^{-2}|\tilde{A}_j(z)|^2$. For the pump and signal intensities, we neglect the transverse spatial intensity profiles (orthogonal to the propagation direction) and instead approximate them via effective mode areas $A_{\text{eff}}(\omega_P) \approx A_{\text{eff}}(\omega_S) \approx 7.5\ \mu m^2$. Further, $\sigma_e$ and $\sigma_a$ are the frequency-dependent effective emission and absorption cross-sections, which we measured for our samples, and $\hbar$ is the reduced Planck constant. The propagation equations of the pump and signal power, which describe the gain along the propagation direction $z$ are

$$\frac{1}{P_P(z)}\frac{dP_P(z)}{dz} = \Gamma_P\Big[\sigma_e(\omega_P)\big(N_{2,a}(z) + N_{2,q}(z)\big)$$
$$-\sigma_a(\omega_P)\big(N_{0,a}(z) + N_{0,q}(z)\big)\Big] - \alpha$$

and

$$\frac{1}{P_{S,j}(z)}\frac{dP_{S,j}(z)}{dz} = \Gamma_S\Big[\sigma_e(\omega_j)\big(N_{1,a}(z) + N_{1,q}(z)\big)$$
$$-\sigma_a(\omega_j)\big(N_{0,a}(z) + N_{0,q}(z)\big)\Big] - \alpha.$$

Here, $\Gamma_S$ and $\Gamma_P$ are the power fractions of signal and pump in the gain medium, respectively, and $\alpha$ describes the propagation loss, which we assume to be the same for both pump and signal. While the pump propagation equation is used directly to model the evolution of $P_P(z)$, the evolution of the signal is modeled via its field amplitude as described above using $g_{S,j}(z) = \frac{1}{P_{S,j}(z)}\frac{dP_{S,j}(z)}{dz}$.

## Data availability

The data shown in the plots have been deposited in the Zenodo database https://zenodo.org/records/12968259; https://doi.org/10.5281/zenodo.12968259.

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

## Acknowledgements

This project has received funding from the Horizon 2020 research and innovation program (FEMTOCHIP, grant agreement No. 965124 (M.G., F.X.K., S.M.G.B., T.H.)), from the European Research Council (ERC) under the EU's Horizon 2020 research and innovation program (STARCHIP, grant agreement No. 853564 (T.H.); CounterLight, grant agreement No. 756966 (P.D.H.)), from the Deutsche Forschungsgemeinschaft (SP2111, contract number 403188360 (F.X.K.)), and through the Helmholtz Young Investigators Group VH-NG-1404 (T.H.); the work was supported through the Maxwell computational resources operated at DESY.

## Author contributions

M.A.G., T.W., and T.H. designed the amplifier chip. M.A.G. developed and performed tests for gain waveguide development. M.A.G. and M.L. designed and built the setup for pulse characterization, performed pulse amplification experiments and analyzed the data. T.V. contributed to the development of pulse input source. M.A.G. developed and performed the numerical simulations. H.F., J.C., and M.G. fabricated the silicon nitride chip. K.W. and S.M.G.B. developed and deposited the gain medium. S.Z., T.B., and P.D.H. supported cladding development. M.S., J.L., N.S., and F.X.K. contributed to the development of gain medium and amplifier. T.H. supervised the work. M.A.G. and T.H. prepared the manuscript with input from all authors.

## Funding

## Competing interests

The authors declare no competing interests.
