## [Peer Review File · Nature Communications]

Femtosecond pulse amplification on a chipEditorial Note: This manuscript has been previously reviewed at another journal that is not operating a transparent peer review scheme. This document only contains reviewer comments and rebuttal letters for versions considered at *Nature Communications*

REVIEWERS' COMMENTS

Reviewer #1 (Remarks to the Author):

I would like to thank the authors for the detailed reply, which addressed most of my concerns. In general, I consider the manuscript to be a good fit for publication in Nature Communications after the revision.

1. I agree that the [14] didn't focus on the detailed design of the waveguide dispersion for high-power pulse amplification and only pointed out the possibility of implementing gain for pulses. Also [27] didn't consider the case of high-peak power pulsed amplification. Although I still hold the view that this work's contribution is incremental as it's based on the same platform, it should have enough importance to warrant publication on Nature Communications.
2. I agree that [26] has limitations on peak power due to the higher nonlinearity effect. The added line does help clarify why we need specially designed pulsed waveguide amplifiers.
3. The authors' argument for the disadvantages of pulse compress after amplification is plausible to me. However, it might be better if the authors could make it clear that the nonlinear effect in the gain waveguide is not the key motivation for performing the chirped amplification when making the analogy to the traditional CPA (around line 87). In my understanding, the large mode area design itself is enough for countering the nonlinearity issue.
4. The authors' clarification of the pump laser is appreciated. Providing the pump power with four diodes through bi-directional pumping and polarization multiplexing is indeed possible (albeit technically challenging).
5. The authors addressed my concern by pointing out the relevant references and analyzed the nonlinearity in 2.4 x 0.8 um silicon nitride waveguides.

Reviewer #2 (Remarks to the Author):

The authors have addressed all the issues raised. I recommend publication in Nature Communications.

Reply to Reviewers

Reviewer report in **black**.

Author reply in **blue**.

Descriptions of revisions are underlined.

Reviewer 1

I would like to thank the authors for the detailed reply, which addressed most of my concerns. In general, I consider the manuscript to be a good fit for publication in Nature Communications after the revision.

1. I agree that the [14] didn't focus on the detailed design of the waveguide dispersion for high-power pulse amplification and only pointed out the possibility of implementing gain for pulses. Also [27] didn't consider the case of high-peak power pulsed amplification. Although I still hold the view that this work's contribution is incremental as it's based on the same platform, it should have enough importance to warrant publication on Nature Communications.
2. I agree that [26] has limitations on peak power due to the higher nonlinearity effect. The added line does help clarify why we need specially designed pulsed waveguide amplifiers.
3. The authors' argument for the disadvantages of pulse compress after amplification is plausible to me. However, it might be better if the authors could make it clear that the nonlinear effect in the gain waveguide is not the key motivation for performing the chirped amplification when making the analogy to the traditional CPA (around line 87). In my understanding, the large mode area design itself is enough for countering the nonlinearity issue.
4. The authors' clarification of the pump laser is appreciated. Providing the pump power with four diodes through bi-directional pumping and polarization multiplexing is indeed possible (albeit technically challenging).
5. The authors addressed my concern by pointing out the relevant references and analyzed the nonlinearity in $2.4 \times 0.8 \text{ um}$ silicon nitride waveguides.

We thank the Reviewer for evaluating our manuscript a second time and for the positive recommendation. We also appreciate the suggestion under point (3) to clarify the key motivation for the chirped amplification. Indeed, the large mode-area waveguides drastically reduce the nonlinearity and up to a certain power level the pre-stretching in a CPA-like scheme would not be strictly needed for those waveguides. However, to create a compact structure, we also need waveguide bends. These bends are necessarily tightly confining and therefore of higher effective nonlinearity. To also avoid nonlinear effects in the bends, the CPA-like scheme is needed. To clarify this point we have made

the following revision:

The central challenge of strong optical nonlinearity inherent to integrated photonics is effectively addressed through tailored large mode-area gain waveguides and the design of the waveguide's dispersion, permitting stable pulse propagation and compression, including in the tightly confining bends.

We are grateful for the valuable comments by the Reviewer throughout the review process and the improvements to the manuscript that they have led to.

Reviewer 2

The authors have addressed all the issues raised. I recommend publication in Nature Communications.

We thank the Reviewer for evaluating our manuscript a second time and for the positive recommendation. We value and appreciate the Reviewer's comments during the review process; they have led to a stronger and more complete manuscript.